

# Short communication: Estimating radiocarbon reservoir effects in Bolivian Amazon freshwater lakes

Asier García-Escárzaga[1,2], Umberto Lombardo[1,2], Patricia M. Bello-Alonso[3,4], José M. Capriles[5], André Colonose[1,2], Kate Dudgeon[2], Carlos D. Simoes[6], Ricardo Fernandes[7,8,9]

[1]Department of Prehistory, Universitat Autònoma de Barcelona, Bellaterra, Spain.
[2]Institute of Environmental Science and Technology (ICTA-UAB), Universitat Autònoma de Barcelona, Bellaterra, Spain.
[3]GEAAT, Group for Archaeology, Antiquity and Territory Studies (GEAAT), Universidade de Vigo, Campus As Lagoas, Ourense, Spain.
[4]TraCEr, Monrepos Archaeological Research Centre and Museum for Human Behavioural Evolution, Leibniz Zentrum für
Archäologie (LEIZA), RGZM, Mainz, Germany.
[5]Department of Anthropology, The Pennsylvania State University, University Park, PA 16802, USA.
[6]Interdisciplinary Center for Archaeology and Evolution of Human Behaviour (ICArEHB), Universidade do Algarve, Faro, Portugal.
[7]Max Planck Institute of Geoanthropology, Department of Archaeology, Jena, Germany.
[8]Department of Bioarchaeology, Faculty of Archaeology, Warsaw, Poland.
[9]Princeton University, Climate Change and History Research Initiative, Princeton, USA.

*Correspondence to*: Asier García-Escárzaga (asier.garcia@uab.cat) and Ricardo Fernandes (fernandes@gea.mpg.de)

**Abstract.** The Llanos de Moxos, in the Bolivian Amazon, preserves a remarkable archaeological record, featuring thousands of forest islands. These anthropogenic sites emerged as a result of activities of the earliest inhabitants of Amazonia during the

Early and Middle Holocene. Excavations conducted to date on the forest islands have revealed that many assemblages contain a high number of ancient freshwater snail remains. In these shell middens, the most represented mollusc taxon, and in most cases the sole one, is *Pomacea* spp., a genus that inhabits inland shallow lakes and wetlands. Although human burials and faunal remains are typically recovered from these sites, their collagen is often not preserved or is of poor quality, and shell carbonates from *Pomacea* shells, along with carbonised plant remains, are often used for [14]C measurements. However, it

remains undetermined if these measurements are subject to radiocarbon reservoir effect (RRE). To determine if a freshwater RRE could affect the age estimations of Amazonian archaeological and other paleoecological deposits, we collected modern coeval *Pomacea* shells and tree leaves from four locations across the Llanos de Moxos area for AMS radiocarbon dating. The radiocarbon results combined with the environmental history of Llanos de Moxos during the Holocene, confirm an absence of significant RREs, and support the continued use of freshwater molluscs as viable material for radiocarbon dating in the region.

**1 Introduction**

Although archaeological research on the earliest human occupations in South America had traditionally prioritised coastal environments (Armesto et al., 2010; Bueno et al., 2013), recent studies have increasingly provided evidence that these populations expanded into the central regions of the continent during the Early Holocene (Lombardo et al., 2013, 2020).





Archaeological research conducted in the Llanos de Moxos, a seasonally inundated tropical savannah in the Bolivian Amazon

(Fig. 1A-B), has revealed that pre-Columbian communities formed artificial mounds known as forest islands since the Early Holocene (Lombardo et al., 2013). These were small forested earthen mounds for which ¹⁴C radiocarbon dates constrain human occupation from approximately 11 ka to 2 ka cal BP (Capriles et al., 2019; Lombardo et al., 2020). Some of these forest islands are composed of shell midden stratigraphic deposits, although not exclusively, and contain a heterogeneous assemblage of archaeological remains (Capriles et al., 2019). Among them, the most common are freshwater mollusc shells of the *Pomacea*

genus (Perry, 1810) (Fig. 1C). Human burials are also frequently found in these sites, along with animal bones and ceramic remains, which are helpful particularly for building relative chronologies. Other evidences, such as wood charcoal or carbonised seeds, may also be encountered at the forest islands, but can be rare in many depositional contexts (Capriles, 2023; Capriles et al., 2019; Lombardo et al., 2013).

The poor preservation of bone collagen (Capriles et al., 2019), along with challenges in dating bioapatite in tropical environments including the difficulties of removing contaminants and diagenesis involving isotopic exchange of dissolved carbon from shells (Cherkinsky, 2009; Fernandes et al., 2013b; Inomata et al., 2022; Zazzo and Saliège, 2011), complicates the radiocarbon dating of bones. Moreover, extensive archaeological excavations are resource-intensive, and much of the available dating evidence comes from coring and auger soil sampling. This methodological limitation significantly restricts

the availability of suitable datable materials, except for *Pomacea* remains, which are abundant, particularly easy to identify and are relatively well preserved. This frequently makes the shells the most viable option for radiocarbon dating of human activities in Southwestern Amazonia and other tropical settings (Lombardo et al., 2013). However, as observed in marine mollusc shells, freshwater specimens may yield radiocarbon ages older than those from coeval terrestrial organic materials (Culleton, 2006; Fernandes et al., 2012, 2013a; Geyh et al., 1997; Inomata et al., 2022; Philippsen, 2013). This ¹⁴C offset

results from the presence of ¹⁴C depleted carbon in water when compared to the contemporaneous atmosphere. This carbon is assimilated by molluscs and incorporated into their shell carbonate structure (Fernandes and Dreves, 2017). Such radiocarbon reservoir effects (RRE), are driven by multiple factors in lacustrine systems. Of particular relevance in our study, is the influx of water enriched with dissolved ancient carbonates (also knowns as hard-water-lake error), transported to lakes via groundwater and runoff (Yu et al., 2007) and carbon contributions from old organic matter (Fernandes et al., 2012, 2013a).

Therefore, determining the magnitude of RREs across time is crucial for the accurate calibration of radiocarbon dates from subfossil shells.

Llanos de Moxos is a noncalcareous region and the river catchment basin in the Andes drains almost exclusively through siliciclastic rock (Gómez Tapias et al., 2019). This has led to a widespread practice of radiocarbon dating lake sediments palaeosols and shells from the Bolivian Amazon without RRE corrections (Carson et al., 2014; Lombardo et al., 2013, 2018;

Whitney et al., 2014). However, RREs may originate also from oxidation of old organic matter (Fernandes et al., 2012, 2013a, 2016). Thus, testing for an absence of freshwater RREs in the Bolivian Amazon remains necessary. Here, we report nine





accelerator mass spectrometry (AMS) radiocarbon dates from *in vivo* collections of *Pomacea* shells and coeval terrestrial plant

samples from four different locations across the Llanos de Moxos area.




**Figure 1: A) Location of the study area in South America. B) Llanos de Moxos in the Bolivian Amazon and the four lakes from which samples were collected. C) A modern specimen of the freshwater *Pomacea* genus from Lago Azul. Maps were created by UL using ArcGIS software, and the photograph of the *Pomacea* shell was taken by AGE.**

## 2 Material and methods

The freshwater golden apple snail genus *Pomacea* (Fig. 1C) is native to South America and has rapidly spread worldwide (Céspedes et al., 2024; Hayes et al., 2008; Seuffert and Martín, 2024). Apple snails of the *Pomacea* genus are part of the *Ampullariidae* family, which includes the largest freshwater snails, reaching up to 17 cm in length (Azmi et al., 2022). *Pomacea* specimens are primarily macrophytophagous, thus preferring floating or submersed plants over emergent ones, although some species, such as *P. canaliculata* also feed on animal matter (Estebenet and Martín, 2002).

In this study modern *Pomacea* spp. specimens were collected alive from four freshwater continental systems located across the Llanos de Moxos region (Fig. 1B). The molluscs were immediately sacrificed to prevent additional calcium carbonate deposition. Additionally, tree leaves of terrestrial trees were collected at the same time near the locations where the molluscs were harvested to provide a reference for atmospheric $^{14}$C values. A total of 5 mollusc shells and 4 tree leaves were subject to AMS radiocarbon measurements at the CIRAM laboratory (France).

The samples of shell were treated with hydrochloric acid (HCl, 1M) to remove any surface contamination. After these pre-treatments, the samples were combusted at 920 °C and transformed into gas using an elemental analyser (EA) (Elementar Vario ISOTOPE Select). During this stage, a first check of the C/N ratio is carried out. The remaining $CO_2$ from the EA outlet was captured by a zeolite trap in an Automated Graphitization Equipment (AGE 3, IonPlus). Subsequently, the gas was released into the designated reactor, where it underwent catalytic conversion into graphite, following the protocol outlined by Vogel *et al.* (Vogel et al., 1984). Measurement was carried out using a Low-Energy Accelerator (LEA, Ionplus AG) and a Single stage accelerator mass spectrometer (SSAMS, NEC) and the conventional radiocarbon age was then calculated following the Stuiver and Polach (1977) convention and considering the $\delta^{13}$C correction for isotopic fractionation, based on the comparison between the concentration measurements of $^{13}$C/$^{12}$C and $^{14}$C/$^{12}$C. The analytical precision of percentage of modern carbon (pMC), which expresses $^{14}$C concentration relative to the $^{14}$C atmospheric level in 1950, is here reported at 1σ. $\delta^{13}$C and $\delta^{15}$N values were obtained independently at CIRAM Laboratory on an isotope ratio mass spectrometry (IRMS), with raw data normalised against international standards (caffein IAEA-600, BCR-657 and IAEA-N-2) (Bohlke et al., 1993; Coplen et al., 2006; Gonfiantini, 1978) and the results expressed per mile (‰) in relation to V-PDB (Vienna Pee Dee Belemnite) for $\delta^{13}$C and AIR (Ambient Inhalable Reservoir) for $\delta^{15}$N.



## 3 Results and discussion

The radiocarbon activity of modern *Pomacea* shells, collected *in vivo* from four lakes, ranged from 100.01±0.37 to
103.03±0.36 pMC, while the terrestrial leaves from neighbouring locations ranged from 100.18±0.37 to 101.55±0.36 pMC
(Table 1). Significant differences in pMC values for terrestrial plant samples collected in 2023 and 2024 reflect the decline in
atmospheric $^{14}$C levels following nuclear weapons testing during the 1950s–1960s (Hua et al., 2022). No statistically significant
differences were observed between the pMC values obtained from coeval lacustrine and terrestrial samples, except for mollusc
LA.1. This specimen exhibited a radiocarbon activity higher than that the reference terrestrial leaf LA.2 and shell LA.3
retrieved from the same lake. The pMC values for LA.2 and LA.3 were not statistically different. The reason for the
discrepancy observed for LA.1 is presently indetermined as we cannot fully exclude localised differences in $^{14}$C carbon sources
available to the two molluscs. The higher $\delta^{13}$C value in LA.1 when compared to LA.2 may be related to a higher consumption
of potentially depleted organic matter by LA.1 (Fernandes and Dreves, 2017).

| Collection site | Collection date | ID Code | Lab Code | Material | pMC | $\delta^{13}$C (‰) | $\delta^{15}$N (‰) |
|---|---|---|---|---|---|---|---|
| Trinidad | Autumn 2023 | CHU.100 | CIRAM-8348 | Shell | 101.54±0.36 | −7.8 | |
| | | CHU.101 | CIRAM-8349 | Tree leaf | 101.08±0.35 | −32.4 | 2.4 |
| Lago Azul | Autumn 2023 | LA.1 | CIRAM-10931 | Shell | 103.03±0.36 | −17.5 | |
| | | LA.3 | CIRAM-13341 | Shell | 101.37±0.35 | −14.6 | |
| | | LA.2 | CIRAM-10932 | Tree leaf | 101.55±0.36 | −30.2 | |
| San Ignacio | Autumn 2024 | SI.1 | CIRAM-12310 | Shell | 100.01±0.37 | −14.9 | |
| | | SI.2 | CIRAM-12311 | Tree leaf | 100.04±0.34 | −29.6 | |
| Santa Rosa de Yacuma | Autumn 2024 | SRO.100 | CIRAM-12312 | Shell | 100.24±0.34 | −14.1 | |
| | | SRO.101 | CIRAM-12313 | Tree leaf | 100.18±0.35 | −32.4 | |

**Table 2: Radiocarbon and stable isotope results for modern mollusc and plant samples measured using AMS and IRMS,
respectively.**

Our results overall confirm the hypothesis that no RRE is expected for modern freshwater *Pomacea* shells within the Bolivian
Amazon. However, we must also consider potential temporal variations in RRE values, as a result of climate change,
hydrological dynamics, and human impacts (Geyh et al., 1997; Philippsen, 2013). Variations in lake sediment sodium
bicarbonate have been linked to evaporation rates (Geyh et al., 1997). However, the absence of carbonate sources in
investigated lakes and climatic stability observed for Llanos de Moxos via palaeoecological records likely exclude climate as
a potential source for RRE temporal variations (Brugger et al., 2016; Mayle et al., 2000). RREs may also reflect geothermal
activity (Ascough et al., 2010), which has not been reported for Llanos de Moxos. Fluvial dynamics in Llanos de Moxos did





change significantly during the Holocene, particularly between 4 ka and 2 ka cal BP, when heightened river activity is recorded (Lombardo et al., 2018). Nevertheless, the Bolivian lowlands and their river catchment areas, with the exception of a carbonate outcrop in the region of Torotoro (Apaéstegui et al., 2018) within the catchment of the Rio Grande river, are mostly devoid of limestone rocks. This allows us to suggest that changes in hydrological dynamics are unlikely to have impacted temporal variations in RRE values for most of the Llanos de Moxos. As for the Rio Grande river, it deposits its sediments in the Santa

Cruz region, where its discharge is significantly reduced as water flows underground (Lombardo, 2016). Rio Grande feeds into the Mamoré river, south of the Llanos de Moxos. However, until approximately 4,000 years ago, the Rio Grande flowed northward, depositing a large sedimentary lobe in southern Llanos de Moxos and likely contributed significantly to sediments that now cover the northeastern part of the region (Lombardo, 2014; Lombardo et al., 2012). Although the carbonate section of the Rio Grande catchment is extremely small compared with non-carbonate rocks, we cannot fully exclude a minor RRE

value in shells dating older than 4,000 years in the eastern part of the Llanos de Moxos.

Humans have impacted the landscapes within our study region to some extent and could have, in theory, locally influenced RRE values. Different studies show that pre-Columbian populations actively managed their surroundings; modifying hydrological conditions to retain water longer into the dry season or drain water more effectively during the wet season

(Lombardo et al., 2025), increasing fire activity (Brugger et al., 2016; Duncan et al., 2021), and constructing geometric earthworks (Carson et al., 2014), among others landscape modifications. However, these human activities primarily took place during the Late Holocene (Erickson, 2000; de Souza et al., 2018; Whitney et al., 2013), following the formation of forest island shell middens, suggesting that their impact during the Early and Middle Holocene, if any, was minimal. Regarding the Late Holocene, these pre-Columbian earthworks influenced the runoff of rain waters, either by speeding up the drainage or via

water retention, on clayish and impermeable soils, with little to no exchange with underground water, as attested by the oxidative patterns of the subsoil (Lombardo et al., 2015). We therefore exclude a notable impact of pre-Columbian human activities on RREs.

In conclusion, while our sample size is still relatively modest, our radiocarbon results, together with an assessment of the

stability of the conditions impacting RRE values, overall confirm the hypothesis that there is an absence of freshwater RREs in the Llanos de Moxos area. This adds support for the reliability of existing radiocarbon chronologies based on $^{14}$C measurements from *Pomacea* shells and incentivises wider use of freshwater molluscs in future radiocarbon dating projects.

**Author contribution**

AGE and UL designed the experiment. AGE, UL, KD, and CDS collected the samples radiocarbon dated in this experiment.
PMBA and AC provided scientific support. RF verified the results, experiments, and other research outputs. AGE prepared



the manuscript with contributions from UL, JMC, and RF. All co-authors have reviewed and edited the final version of the manuscript.

## Acknowledgements

We would like to thank Maicol Apomaita for collecting the modern mollusc shells used for radiocarbon dating in this investigation and to Juan Pablo Llapiz for his invaluable support during our stay at Lago Azul.

## Funding sources

This research was carried out as part of the I+D+i project PID2022-138350OA-I00, funded by MICIU/AEI/10.13039/501100011033 and by ERDF/EU. This work was also supported by the ERC Consolidator project DEMODRIVERS funded by the European Research Council (ERC) (ID: 101043738; doi: 10.3030/101043738). During the 165 development of this research AGE has also been funded by the European Commission through a Marie Skłodowska Curie Action – Postdoctoral Fellowship (NEARCOAST; ID: 101064225; doi: 10.3030/101064225). PBA is currently supported by Post-Doc Xunta de Galicia Grant (ID: ED481B-2022/079). This work also contributes to the ICTA-UAB "María de Maeztu" Programme for Units of Excellence of the Spanish Ministry of Science, Innovation and Universities (CEX2024-001506-M) and to the EarlyFoods project, which has received funding from the Agència de Gestió d'Ajuts Universitaris i de Recerca de 170 Catalunya (SGR-Cat-2021, 00527).

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
