# Peer review of "Short communication: Estimating radiocarbon reservoir effects in Bolivian Amazon freshwater lakes"

_EGUsphere, 2025_

## Author Response (AR2)

**Point-by-Point Response to Editor**

We would like to thank the editor and reviewers for reading the new version of manuscript and we thank all for their additional comments and suggestions. We have responded to these on a point-by-point basis below and modified our manuscript (see separate Word file). Changes carried out following reviewers and editor comments/suggestions are highlighted in red and green colours, respectively.

**Line 25: It would be better to use FRE from the start and throughout the whole manuscript.**

We employed the expression RRE for consistency with our own previous manuscripts and others (e.g., Alves et al. 2025). Our preference goes for RRE since it includes "Radiocarbon" within it. However, we acknowledge that FRE (Freshwater reservoir effects) is also sometimes employed, and we modified a sentence in the introduction: "Such radiocarbon reservoir effects (RRE) in lacustrine systems, also known as freshwater reservoir effects (FRE), are driven by multiple factors."

**Line 30: Responses of authors in the discussion suggest that Alves et al. 2025 is known to the authors. Please refer to that paper in the introduction.**

The reference has been added as follows: "However, as observed in marine mollusc shells, freshwater specimens may yield radiocarbon ages older than those from coeval terrestrial organic materials (Alves et al., 2025; Culleton, 2006; Fernandes et al., 2012, 2013a; Geyh et al., 1997; Inomata et al., 2022; Philippsen, 2013).".

Lines 85/86: Comments to the review analysis at the Vilnius laboratory are noted, but not here. Please make the distinction clear. I understand that CIRAM has an LEA machine. Is this correct? And who burned the shells in EA? I think both labs have the AGE system.

Following previous reviewer's comments we included in the text: "A total of five mollusc shells and four tree leaves were subject to sample pre-treatment, combustion, and graphitization at the CIRAM laboratory (Martillac, France) while AMS measurements were carried out at BARNAS mass spectrometry (Vilnius, Lithuania).". Combustion and graphitization stages were performed using an EA and an AGE located at CIRAM. Likewise, we also clarified that LEA refers to the AMS model, which is located at BARNAS.

To improve clarity, we have modified the sentences to read:

"Following chemical pre-treatments, both shell and leaf samples (weights ranging from 3.5 and 15.4 mg) were combusted at 920 °C using an elemental analyser (EA) (Elementar, Vario ISOTOPE Select) at the CIRAM laboratory.".

"Measurements of the radiocarbon content for both shell and leaf samples were carried out using a 50 kV accelerator mass spectrometer Low-Energy Accelerator (LEA, IonPlus AG) at the BARNAS laboratory.".

"Measurements of stable carbon isotope ratios for leaf and shell samples were carried out at the CIRAM laboratory using c. 10% of the EA produced CO2 (see previous paragraph) which flowed into a coupled isotope ratio mass spectrometry (IRMS) (Elementar, isoprime precision).".

Lines 93/94 (about the Vogel et al. 1984 reference): indeed, it is correct that the graphitization follows this original publication. The AGE innovation involves combustion in EA and the transfer of CO2 using a He stream. But the chemical reaction is Vogel et al., 1984.

We have double-checked this with the CIRAM laboratory, and they have confirmed the same to us.

Line 97: It is not a correct expression about the levels of 14C in 1950. Radiocarbon ages are calculated back to 1950, but the 14C level in the atmosphere had already changed by the Suess effect. Therefore, the pre-industrial value was used. Here, it is better to say normalized to the standard in 1950. For this, numerous publications explain it. Here is one: Primer: <a href="https://rdcu.be/ejaRp">https://rdcu.be/ejaRp</a>.

We grateful for noting this and we have modified the sentence accordingly: "The analytical precision of the Fraction Modern (F14C), which expresses 14C concentration normalised to the standard 14C atmospheric level in 1950...".

Line 113 (table 1): This table does not show Vilnius samples. In addition, as discussed in the comments to the review, the value of LA.3 is said to be lower than that of LA.1. It is not; it is less negative, which means it's higher. Are these  $\delta$ 13C all IRMS values? AMS provides these numbers as well, and they can be influenced by sample preparation (sample size) and the AMS itself. So please give a clear description of the columns in Table 2, as well as the info about the amount of Carbon used for analysis.

We do not understand what the editor means by 'This table does not show Vilnius samples.' Stable carbon isotope ratio values for all samples were measured using an IRMS at the CIRAM laboratory in France. A new sentence was added following the reviewer's comment to improve the clarity of the methods section: "Measurements of stable carbon isotope ratios for leaf and shell samples were carried out at the CIRAM laboratory using c. 10% of the EA produced CO2 (see previous paragraph) which flowed into a coupled isotope ratio mass spectrometry (IRMS) (Elementar, isoprime precision).".

Yes, we made a mistake in our original version regarding d13C values obtained from LA.1 and LA.3 samples, and we corrected this as the reviewer indicated: "The higher  $\delta^{13}$ C value in LA.3, when compared to LA.1, may be related to a higher incorporation of carbon sourced from organic matter by LA.1 (mollusc shell LA.1 also showed the lowest  $\delta^{13}$ C value of all shell samples)…".

Information about the amount of sample introduced into the EA for subsequent graphitization through AGE and IRMS analysis has been included in the methods section: "Following chemical pre-treatments, both shell and leaf samples (weights ranging from 3.5

and 15.4 mg) were combusted at 920 °C using an elemental analyser (EA) (Elementar, Vario ISOTOPE Select) at the CIRAM laboratory. [...] The  $CO_2$  emerging from the EA was split with c. 90% of this captured by a zeolite trap in an Automated Graphitization Equipment (AGE) (IonPlus AG, AGE 3)..." and "Measurements of stable carbon isotope ratios for leaf and shell samples were carried out at the CIRAM laboratory using c. 10% of the EA produced  $CO_2$  (see previous paragraph) which flowed into a coupled isotope ratio mass spectrometry (IRMS)...".

A new table 1 has been added to include the information asked by the editor. We specify that d13C values were obtained using an IRMS.

| Collection site         | Collection date | ID Code | Lab Code    | Material  | F 14 C | χ2 test results
df = 1 (5%, 3.8) | δ 13 C (‰)
(by IRMS) |
|-------------------------|-----------------|---------|-------------|-----------|-------------------|-------------------------------------|------------------------------------|
| Trinidad                | Autumn 2023     | CHU.100 | CIRAM-8348  | Shell     | 1.0154±0.0036     | T = 0.8                             | -7.8                               |
|                         |                 | CHU.101 | CIRAM-8349  | Tree leaf | 1.0108±0.0035     |                                     | -32.4                              |
| Lago Azul               | Autumn 2023     | LA.1    | CIRAM-10931 | Shell     | 1.0303±0.0036     | T = 8.5                             | -17.5                              |
|                         |                 | LA.3    | CIRAM-13341 | Shell     | 1.0137±0.0035     | T = 0.1                             | -14.6                              |
|                         |                 | LA.2    | CIRAM-10932 | Tree leaf | 1.0155±0.0036     |                                     | -30.2                              |
| San Ignacio             | Autumn 2024     | SI.1    | CIRAM-12310 | Shell     | 1.0001±0.0037     | T = 0.0                             | -14.9                              |
|                         |                 | SI.2    | CIRAM-12311 | Tree leaf | 1.0004±0.0034     |                                     | -29.6                              |
| Santa Rosa de
Yacuma | Autumn 2024     | SRO.100 | CIRAM-12312 | Shell     | 1.0024±0.0034     | T = 0.0                             | -14.1                              |
|                         |                 | SRO.101 | CIRAM-12313 | Tree leaf | 1.0018±0.0035     |                                     | -32.4                              |
|                         |                 |         |             |           |                   |                                     |                                    |

It is still confusing to have CIRAM numbers as lab numbers in Table 1, and the caption says AMS. If you have AMS #s add them to the table. The purpose of publishing lab numbers is to facilitate tracking the results in the future.

The new Table 1 already includes the lab code. We have double-checked this with CIRAM, and they confirmed that the code is the same for both the CIRAM and BARNAS labs.

It is not typical to use EA. It is possible, but not typical, so please get in touch with CIRAM and add the line explaining how they did that.

CIRAM has provided additional information regarding the combustion processes. We've incorporated this new information:

Samples were weighed into tin capsules and combusted in the EA using oxygen mixed with helium carrier gas ( $O_2$  dosing time: 100 seconds). The carbonates were combusted following a similar procedure, but these were ground into powder before being weighed into tin capsules and subject to a longer combustion time ( $O_2$  dosing time: 120 seconds).